# ASPIRO: Any-shot Structured Parsing-error-Induced ReprOmpting for Consistent Data-to-Text Generation

**Martin Vejvar**
Graduate School of Engineering Science
Yokohama National University
vejvar-martin-km@ynu.jp

**Yasutaka Fujimoto**
Faculty of Engineering
Yokohama National University
fujimoto@ynu.ac.jp

## Abstract

We present ASPIRO, an approach for structured data verbalisation into short template sentences in zero to few-shot settings. Unlike previous methods, our approach prompts large language models (LLMs) to directly produce entity-agnostic templates, rather than relying on LLMs to faithfully copy the given example entities, or validating/crafting the templates manually. We incorporate LLM re-prompting, triggered by algorithmic parsing checks, as well as the PARENT metric induced consistency validation to identify and rectify template generation problems in real-time. ASPIRO, compared to direct LLM output, averages 66% parsing error rate reduction in generated verbalisations of RDF triples on the DART dataset. Our best 5-shot text-davinci-003 setup, scoring BLEU of 50.62, METEOR of 45.16, BLEURT of 0.82, NUBIA of 0.87, and PARENT of 0.8962 on the Rel2Text dataset, competes effectively with recent fine-tuned pre-trained language models.[1]

## 1 Introduction

Data-to-text task (Reiter, 1996) aims to build a faithful natural language interpretation of structured data such as relational tables or Resource Description Framework (RDF) triples (Miller, 2001). However, without proper context, the given structured data may not sufficiently represent the relationships between entities, leading to ambiguity (Dušek et al., 2019). To battle this, some works rely on fine-tuning pre-trained language models (PLMs) on task-specific datasets in supervised or semi-supervised ways (Ke et al., 2021; Agarwal et al., 2021), but the domain of the resulting system is limited and requires well-labelled training data (Keymanesh et al., 2022). In contrast to fine-tuning, Kasner and Dusek (2022) prove that zero-shot neural systems are a possible solution, where in-domain data is introduced via simple human-crafted templates for each unique relation in the

knowledge graph. Xiang et al. (2022) nullify the requirements for human labelling entirely by utilising GPT3-davinci (Brown et al., 2020), a large language model (LLM) with broad general knowledge, to disambiguate RDF triples into short sentences and automatically parse them into reusable sentence templates as an alternative to human-crafted templates. In this paper we introduce ASPIRO, a robust $N$-shot variant of the data disambiguation step presented by Xiang et al. (2022) and a promising alternative to fine-tuning PLMs for crafting RDF verbalisations (Kasner et al., 2023). At its core, ASPIRO uses simple rules to algorithmically flag errors in the templates (such as missing subject, multiple objects, etc.) and re-prompt the LLM until all errors are alleviated or maximum ($N$) retries have been reached. We evaluate changes in automated metrics and reduction of parsing errors in different configurations of ASPIRO on DART (Nan et al., 2021) and Rel2Text (Kasner et al., 2023) and compare the original RDF verbalisation prompt used by Xiang et al. (2022) with our prompt focused on enforcing structured json output with intermediate fields as guidelines.

## 2 Related Work

**Single triple verbalisation:** Mainly leveraged for reducing ambiguity in structured data before a specific D2T task (Laha et al., 2019; Dušek and Kasner, 2020; Xiang et al., 2022) as well as transforming inputs to be better suited for existing NLG models (Gupta et al., 2020; Kasner and Dusek, 2022; Xiang et al., 2022), verbalisation templates fall into three main categories:

1) human-crafted (Kale and Rastogi, 2020; Kasner and Dusek, 2022)
2) rule-based (Laha et al., 2019; Gupta et al., 2020)
3) neural model-based (Xiang et al., 2022; Kasner et al., 2023)

ASPIRO combines aspects of both 2) and 3).

---

[1]code available at github.com/vejvarm/ASPIRO.

**Delexicalization:** Einolghozati et al. (2020) and Heidari et al. (2021) find that without delexicalization, generative models can produce incomplete representations of the entities and concepts in the structured data verbalisations, leading to misinterpretation and failures in production. Our JSON structured prompt (§G.2) enforces the LLM to directly produce named-entity agnostic templates.

**0-shot to $N$-shot:** Our work is heavily inspired and builds upon the disambiguation step from Xiang et al. (2022), which is equivalent to 0-shot setting for our $N$-shot Generator. We also use their prompt (§G.1) as baseline against our JSON prompt (§G.2).

**Refining LLM outputs:** Madaan et al. (2023) and Shinn et al. (2023) show that iterative prompting and chain-of-thought reasoning can significantly improve the outputs of LLMs. We lean on their findings in designing our ASPIRO pipeline. However, back and forth prompting of LLMs can be expensive, which we counterweight by using our Rule-based parser (§3.1) and the PARENT (Dhingra et al., 2019) F1 score (§3.2) as cost-efficient gateways to decide if additional prompting is necessary.

## 3 Methods

The proposed method (ASPIRO) revolves around the conversion of structured data samples into verbalisation templates using a two-stage pipeline: $N$-**shot Generator** (§3.1) and **Consistency Validator** (§3.2). The pipeline processes structured data samples, wherein each sample comprises of one or more RDF triples which share the same relation. ASPIRO (see Figure 1) starts with an initial prompt to verbally articulate the structured data. This is equivalent to prompting a single LLM directly. If the zeroth attempt isn't accurate, it will retry a maximum of $N$ times, refining the previous completion based on parsing errors (§3.1.2). Subsequently, the outputs are validated for consistency, ensuring faithful and reliable verbalisations. We explain the individual stages and their sub-modules in the sections below. Refer to Figure 1 for full pipeline and terminology on general input. **Step-by-step flow** of the pipeline and example on specific input are provided in section §3.3 and Figure 2 respectively.

### 3.1 $N$-shot Generator

$N$-shot Generator further fractures into an LLM stack and a Rule-based parser. The LLM Stack is tasked with generating verbalisation attempts based on given initial prompt (§G.1). It does so with the help of the Rule-based parser. This parser checks the generated completions for structural accuracy, ensuring they adhere to expected patterns.

#### 3.1.1 LLM Stack

The LLM stack is a sequence of $N + 1$ LLMs, indexed from 0 to $N$. $\mathcal{L}_0$ is responsible for the initial completion and each further retry shot, initiated by the Rule-based parser (§3.1.2), increments the index by 1. Each $L_n$ is instantiated separately and does not have to be the same model. Equation (1) shows the single completion for structured input sample $x$ at shot $n$.

$$y_n = \mathcal{L}_n(\mathcal{T}(x)) \qquad (1)$$

where $\mathcal{T}$ is a given prompt and can be either $\mathcal{T}_I$ (initial) or $\mathcal{T}_R$ (retry).

#### 3.1.2 Rule-based parser

A purely algorithmic module, which validates $y_n$ against a set of conditions $\{\mathcal{C}\}$ one by one. If $y_n$ does not pass the condition $\mathcal{C}_i$, a respective parsing error is logged into set $\mathcal{E}_n$. The aggregated rules for each given completion are formally given below (see §A for detailed Python implementation).

$\mathcal{C}_0$ ... has exactly one '<subject>' substring.
$\mathcal{C}_1$ ... has exactly one '<object>' substring.
$\mathcal{C}_2$ ... has no other '<...>' substrings.

If the parser identifies any error in the structure, the next LLM in the LLM stack is re-prompted with Retry prompt (§G.3) to generate new completion.

### 3.2 Consistency Validator

Even if the outputs from the $N$-shot Generator adhere to the structural patterns, they might still contain inaccuracies, such as hallucinated content. This module assesses the quality of the verbalisations, using the PARENT statistical metric (Dhingra et al., 2019). If PARENT F1 score is too low, the module will utilise an LLM with specialised Consistency prompt (§G.4) to improve the sentence.

#### 3.2.1 PARENT$_{F1}$ threshold

To gauge the quality of the completion $y_n$ from $N$-shot Generator, we set a minimal threshold ($\mu$) for the PARENT score of $y_n$. The score is calculated

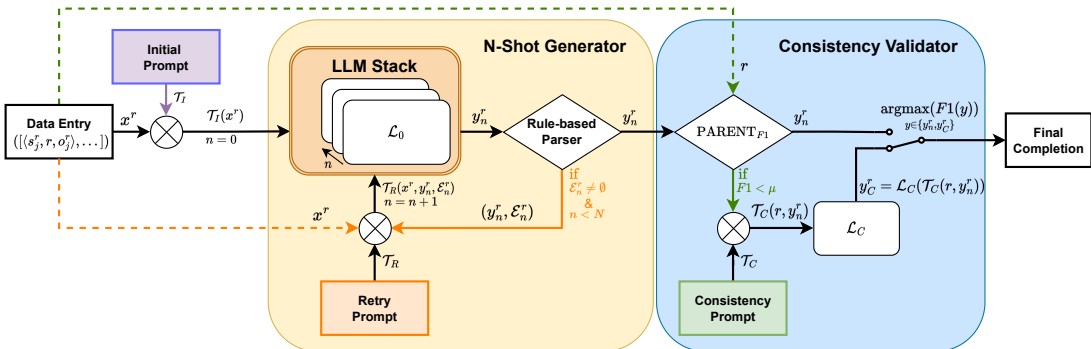

Figure 1: ASPIRO pipeline for general input sample $x^r \in X$.

using eq. (3) against artificially constructed table and reference.

First, we construct the respective hypothesis, table and reference entries:

$$h = y_n.\texttt{replace}([s, o], e)$$
$$t = \langle e, r.\texttt{split}("\ "), e \rangle \quad (2)$$
$$\rho = r$$

where "<subject>" and "<object>" are replaced with "<entity>" to prevent penalising order discrepancy between hypothesis and table.

We then calculate the PARENT F1 score using equation (3).

$$F1(y_n) = \texttt{PARENT}(h, \rho, t) \quad (3)$$

### 3.2.2 Consistency LLM

If the calculated PARENT score from §3.2.1 is not sufficient, we call another LLM with prompt $\mathcal{T}_C$ as in eq. (4).

$$y_C = \mathcal{L}_C(\mathcal{T}_C(r, y_n)) \quad (4)$$

The prompt $\mathcal{T}_C$ is designed to guide $\mathcal{L}_C$ to identify problems with the given completion, provide advice how to fix it and subsequently produce fixed completion in a structured json output. See §G.4 for full version of the prompt.

### 3.3 Stepwise Pipeline Formulation

Given a dataset of structured data samples $\{x^r\}_{r \in \mathcal{R}}$, where $x^r = \{x_1^r, x_2^r, ..., x_m^r\}$ and $x_j^r$ is a single RDF triple $x_j^r = \langle s_j^r, r, o_j^r \rangle$ with relation $r \in \mathcal{R}$, the pipeline for one $x^r$ is as follows:

**Step 0** Set $n = 0$ and $\mathcal{T}_0^r = \mathcal{T}_I(x^r)$.

**Step 1** Calculate $y_n^r$ using eq. (1).

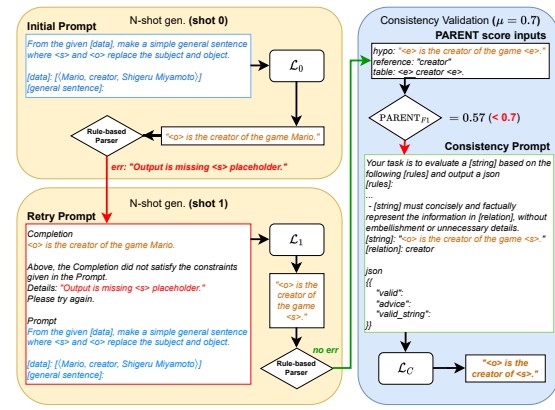

Figure 2: Example flow of ASPIRO pipeline with input sample $x^r = [\langle \text{Mario, creator, Shigeru Miyamoto} \rangle]$

**Step 2** Use §3.1.2 to validate $y_n^r$ against all conditions $\mathcal{C}$. If errors ($\mathcal{E}_n^r$) are found, run equation (5) and return to **Step 1**. Otherwise go to **Step 3**.

$$\mathcal{T}_{n+1}^r = \mathcal{T}_R(x^r, y_n^r, \mathcal{E}_n^r)$$
$$n = n + 1 \quad (5)$$

**Step 3** Use §3.2.1 and calculate $F1(y_n^r)$ via eq. (3). If the calculated $F1$ score is lower than our chosen threshold $0 \le \mu \le 1$, continue to **Step 4**. Otherwise, output current $y_n^r$ as the final completion $y^r$.

**Step 4** Use §3.2.2 to get revised completion $y_C^r$.

**Step 5** Compute $F1$ scores of $y_n^r$ and $y_C^r$ using eq. (3) and take the completion with higher score via eq. (6) to produce the final completion $y^r$.

$$y^r = \underset{y \in \{y_n^r, y_C^r\}}{\texttt{argmax}}(F1(y)) \quad (6)$$

## 4 Experiments

The following sections show results on several setups of ASPIRO. In section §4.1 we compare auto-

Table 1: Average values from 5 independent runs of 4 ASPIRO configurations on Rel2Text test set using automatic metrics (desc. §C), compared with Kasner et al. (2023)'s BART-BASE models, fine-tuned on full (**full-**) or only X (**fewshot-X**) examples from Rel2Text training set. See Table 2 for models and Table 7 for standard deviations.

| all test samples (616) | BLEU ↑ | METEOR ↑ | BLEURT ↑ | NB ↑ | SS ↑ | C (%) ↓ | N (%) ↓ | E (%) ↑ | PARENT$_{F1}$ ↑ |
|---|---|---|---|---|---|---|---|---|---|
| (**Kasner et al., 2023**) full-rel2text | **52.54** | 44.86 | 0.54 | **0.88** | 4.72 | 3.50 | **4.65** | **91.85** | - |
| (**Kasner et al., 2023**) fewshot-25 | 31.13 | 35.52 | -0.02 | 0.65 | 3.94 | 8.35 | 27.26 | 64.39 | - |
| (**Kasner et al., 2023**) fewshot-200 | 48.67 | 43.34 | 0.44 | 0.83 | 4.58 | 5.40 | 9.03 | 85.57 | - |
| (A) G3.5 | 51.22 | 44.73 | 0.82 | 0.87 | 4.71 | 4.88 | 7.34 | 87.77 | 0.8883 |
| (A) 5xG3.5 (G3.5) | 51.40 | 44.94 | 0.82 | 0.87 | 4.72 | 3.65 | 7.98 | 88.38 | 0.8895 |
| (J) G3.5 | 50.63 | 45.13 | 0.82 | 0.87 | **4.79** | 1.60 | 7.36 | 91.04 | **0.8963** |
| (J) 5xG3.5 (G3.5) | 50.62 | **45.16** | 0.82 | 0.87 | 4.79 | **1.54** | 7.25 | 91.21 | 0.8962 |

Table 2: LLM variants used in our experiments.

| label | full name | model family | original publication |
|---|---|---|---|
| G3 | davinci | GPT3 | (Winata et al., 2021) |
| G3.5 | text-davinci-003 | InstructGPT | (Ouyang et al., 2022) |
| G3.5T | gpt-3.5-turbo-0301 | ChatGPT | (OpenAI, 2022) |
| G4 | gpt-4-0314 | GPT4 | (OpenAI, 2023) |

matic metrics on Rel2Text test set (§D.3) with Kasner et al. (2023)'s fine-tuned BART-BASE models. In section §4.2 we report on the number of parsing errors tagged by our Rule-based parser (§3.1) on both DART (§D.1) and Rel2Text (§D.3) datasets. In §4.3 we also provide brief ablation study of CV.

**Setup:** For $N$-shot generator (§3.1), $\mathcal{L}_0$ marks initial model choice and $Nx\mathcal{L}_n$ max $N$ retry shots using model $\mathcal{L}_n$. We limit our experiments to $\mathcal{L}_n$ being same for all $N$ shots. For Consistency Validator (§3.2), we set $\mu = 0.7$ and only use it in some setups (marked by $\mathcal{L}_C$ in brackets). For reference on LLMs used as $\mathcal{L}$ in ASPIRO setups, see Tab. 2.

**Prompts:** While Retry prompt $\mathcal{T}_R$ (§G.3) and Consistency prompt $\mathcal{T}_C$ (§G.4) are constant across all our experiments, we compare two variants of the Initial prompt $\mathcal{T}_I$:

(**A**) ASDOT: proposed by Xiang et al. (2022) in their Data Disambiguation step to produce short sentence representation of a given triple. (full form §G.1)

(**J**) JSON: our proposed prompt, which enforces json-like output with auxiliary fields to guide the creation of named-entity agnostic templates directly. (full form §G.2)

### 4.1 Automatic Metrics

We evaluate Automatic metrics on Rel2Text test set (§D.3) with 4 ASPIRO setups (see Table 1 for 5 run averages; Table 7 for standard deviations).

ASPIRO outperforms Kasner et al. (2023)'s fewshot-fine-tuned PLMs on all metrics and is

competitive to the full-training-set-fine-tuned full-rel2text model with ca 1-2 point reduction in BLEU, but 28 % points increase in BLEURT. This implies higher semantic similarity, however **S**emantic **S**imilarity sub-score of **NUBIA** only shows small increments. Despite the overall **NB** score being same for all ASPIRO setups, the sub-metrics of NUBIA show steady improvement between our models. Most noticeable change is in the **C**ontradiction percentage, which the 5-shot setting improves by ca 1.2 % points and further 2 % points by introducing JSON prompt, suggesting higher capacity to disambiguate the correct direction of relation between subject and object entities in the input triples. PARENT F1 score slightly favours the JSON prompted setups of ASPIRO, but only by ca 0.6 % points.

**Additional experiments:** For metric results and discussion on DART, see appendix §E.1. For full experiment results with fine-tuned pre-trained language models refer to (Kasner et al., 2023).

Table 3: **DART** dataset counts of templates with parsing errors. **RR %**: error Rate Reduction percentage of best $N$-shot setup (bold) vs 0shot model.

| Initial run | | | $N$-shot Gen Setup [$Nx\mathcal{L}_n$ (+$\mathcal{L}_C$)] | | | | |
|---|---|---|---|---|---|---|---|
| $\mathcal{T}_I$ | $\mathcal{L}_0$ | 0x | 1xG3.5T | 1xG3.5 | 1xG4 | 5xG3.5T (+G3.5T) | RR % |
| (A) | G3 | 719 | 467 | 490 | 351 | **239** (227) | 66.76 (68.43) |
| (A) | G3.5T | 996 | 727 | 833 | 784 | **497** | 50.10 |
| (J) | G3.5T | 194 | 166 | 156 | **58** | 80 | 70.10 |
| (J) | G3.5 | 94 | 75 | 59 | **24** | 25 | 74.47 |

Table 4: **Rel2Text** counts of templates with errors.

| Initial run | | | $N$-shot Gen Setup [$Nx\mathcal{L}_n$ (+$\mathcal{L}_C$)] | | | | |
|---|---|---|---|---|---|---|---|
| $\mathcal{T}_I$ | $\mathcal{L}_0$ | 0x | 1xG3.5T | 1xG3.5 | 1xG4 | 5xG3.5T (+G3.5T) | 5xG3.5 (+G3.5) |
| (A) | G3.5 | 21 | 20 | 21 | 19 | **11** (9) | 21 (14) |
| (J) | G3.5 | 3 | 3 | 3 | 2 | **1** (1) | 2 (2) |

Table 5: Automatic metrics without and with consistency validation (CV) for Rel2Text test set.

| Rel2Text test samples (616) | BLEU ↑ | METEOR ↑ | BLEURT ↑ | NB ↑ | SS ↑ | C (%) ↓ | N (%) ↓ | E (%) ↑ | PARENT$_{F1}$ ↑ |
|---|---|---|---|---|---|---|---|---|---|
| 5xG3.5T | 50.88 | 44.97 | 0.8215 | 0.8725 | 4.73 | 3.36 | **8.04** | 88.59 | 0.8910 |
| 5xG3.5T (CV G3.5T) | 50.89 | 44.95 | 0.8213 | 0.8718 | 4.73 | **3.02** | 8.34 | 88.64 | 0.8908 |
| 5xG3 | 50.96 | 44.69 | 0.8200 | 0.8680 | 4.71 | 4.92 | **7.59** | 87.48 | 0.8879 |
| 5xG3 (CV G3) | **51.32** | 44.77 | 0.8200 | 0.8702 | 4.72 | **3.35** | 7.70 | **88.94** | 0.8895 |

## 4.2 Parsing Errors

Parsing error analysis does not require specific references from the dataset. After ASPIRO produces the verbalisation templates ($y^r$), we run them through our Rule-based parser (§3.1) to flag and count the number of errors. As source data ($X$), similar to (Xiang et al., 2022), we collect at most 2 triple examples for each unique relation in the dataset and use them to prompt our pipeline.

**Parsing error counts:** For **DART** (Table 3) we use the full dataset (§D.1), producing 4299 unique template sentences in each experiment run. In **Rel2Text** (Table 4) we only use the test split (§D.3) with 226 unique relations and G3.5 (T2) as base model with either (**A**)SDOT or (**J**)SON prompts and different $N$-shot Generator setups. For Rel2Text, we don't provide RR % as the reduction is evident from counts.

**Discussion:** Introducing $N$-shot Generator (§3.1) shows significant reduction in parsing error counts (Tables 3 and 4) even with $N = 1$. In the 1 retry shot setting, GPT4 (**G4**) is most effective at reducing parsing errors. However, if we introduce up to 5 retry shots, we can see that gpt-3.5-turbo (**G3.5T**) reduces parsing errors further. The exception is (**J**)SON prompt on DART where G4 keeps the lead. Interestingly, while text-davinci-003 (**G3.5**) performs well as 0-shot model, it generally performs worse than G3.5T in $N$-shot settings, contrasted again on DART by **J** prompt. It is also evident that **J** prompt provides more robust 0-shot baseline compared to (**A**)SDOT prompt. The values in parentheses reveal that including Consistency Validation yields only slight reduction in error count.

## 4.3 Ablation of Consistency Validator

To investigate the efficacy of Consistency Validator, we conduct a brief ablation study on Rel2Text test set (§D.3). For statistical metrics (Table 5), CV provides only marginal gains. This effect may be attributed to the improvement of **C**ontradiction score and degradation of **N**eutrality score, implying that CV moves the templates closer to general state-

ments with less informational value. Conversely, parsing errors (Table 6) are reduced notably by CV, with counts decreasing from 12 to 10 and 23 to 16.

Table 6: Individual error counts ($|\mathcal{E}|$) without and with CV on Rel2Text test set with (A)SDOT prompt.

| NxLLM (CV) | Total $|\mathcal{E}|$ | $|$Templates$|$ w/ $\mathcal{E}$ | Missing SUBJ. | Missing OBJ. |
|---|---|---|---|---|
| 5xG3.5T | 12 | 11 | 2 | 10 |
| 5xG3.5T (G3.5T) | 10 | **9** | 2 | 8 |
| 5xG3 | 23 | 21 | 5 | 18 |
| 5xG3 (G3) | 16 | **14** | 3 | 13 |

## 5 Conclusion

We proposed and evaluated ASPIRO, a general domain-agnostic pipeline for verbalisation of single triple data entries to short template sentences, utilising rule-based re-prompting of LLMs. The pipeline comprises of $N$-shot Generator (§3.1) and Consistency Validator (§3.2). We show that AS-PIRO compares to fine-tuned pre-trained language models' automatic scores on the Rel2Text test set (§4.1) and significantly reduces the parsing error count in 0-shot outputs of LLMs (§4.2). The ablation study (§4.3) revealed that Consistency Validator of ASPIRO further reduces error counts, but does not significantly affect automatic metrics.

## Limitations

**Operational costs:** When contrasted with 0-shot setting, ASPIRO significantly escalates the operational costs (see appendix §F) due to the repeated calls of the $N$-shot Generator and the lengthy Consistency prompt (§G.4) associated with the Consistency Validator (§3.2). Following the brief ablation study of CV (§4.3) and the cost analysis, it remains debatable whether the performance of the Consistency Validator reported in this paper justifies the additional expense incurred in prompting the LLM for the flagged examples.

**Isolated triples:** Generating verbalisations from single isolated triples doesn't account for situations where context from other triples is necessary to fully interpret the final natural language verbalisation. As exemplified by the DART dataset, contextual integration is significant and should be explored further.

**Backup template:** In instances where the parsing of the <subject> and <object> within the generated completion of the LLM proved unsuccessful, Xiang et al. (2022) introduced a general backup template as fallback. In our research, we did not use any backup templates and did not investigate their potential impact on automated metric scores. Nonetheless, it's important to acknowledge that within a production environment, the incorporation of a backup template is a fundamental necessity, warranting further assessment of its effects.

**Direction of relation:** The capacity to accurately discern the correct direction of the relation between subject and object is a notable feature of Data-to-text systems. In our experiments, we report on contradiction statistic (C %), which can roughly translate to measure this ability. Although ASPIRO generally shows to improve on this statistic, there are no specific guardrails to validate the ambiguity other than the general knowledge of the LLM itself.

**Variance of experiment runs:** Due to the substantial expenses associated with prompting large language models (LLMs) and the considerable size of the DART dataset, each experiment on DART was conducted only once. The same is true for Rel2Text parsing error analysis in Table 4. It should be noted that, although the temperature parameter was uniformly set to 0 for all the employed LLMs, the underlying generative process remains reliant on maximum likelihood estimation, which inherently leaves room for potential variation errors in our experimental results.

## Ethics Statement

In the course of this research, we have employed various Generative Pre-trained Transformer models, including GPT3 davinci, InstructGPT text-davinci-003 and gpt-3.5-turbo-0301, and gpt-4-0314, each demonstrating inherent biases as outlined in their respective publications, which are listed in Table 2. These biases often favour popular opinions and can lead to a distortion in the model's outputs. This reflects the models' training on large-scale internet text, which is not entirely neutral and contains biased or skewed perspectives. We acknowledge this limitation and highlight that despite implementing a pipeline designed to minimise the inclusion of unnecessary and irrelevant information, the potential for biased outcomes cannot be entirely eliminated.

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

Table 7: Standard deviations of results from Table 1. The values for pre-trained language models in first three rows are copied for reference from Kasner et al. (2023)'s Table 6.

| all Rel2Text test samples (616) | BLEU↑ | METEOR↑ | BLEURT↑ | NB↑ | SS↑ | C (%)↓ | N (%)↓ | E (%)↑ | PARENT$_{F1}$↑ |
|---|---|---|---|---|---|---|---|---|---|
| (Kasner et al., 2023) full-rel2text | 0.60 | 0.30 | 0.01 | 0.01 | 0.02 | 0.41 | 0.38 | 0.65 | - |
| (Kasner et al., 2023) fewshot-25 | 1.60 | 1.18 | 0.05 | 0.03 | 0.14 | 1.19 | 2.67 | 3.58 | - |
| (Kasner et al., 2023) fewshot-200 | 0.80 | 0.35 | 0.02 | 0.01 | 0.02 | 0.60 | 1.37 | 1.25 | - |
| (A) G3.5 | 0.10 | 0.02 | 0.00 | 0.00 | 0.00 | 0.00 | 0.10 | 0.09 | 0.0002 |
| (A) 5xG3.5 (G3.5) | 0.63 | 0.11 | 0.00 | 0.00 | 0.01 | 0.16 | 0.12 | 0.14 | 0.0005 |
| (J) G3.5 | 0.13 | 0.06 | 0.00 | 0.00 | 0.00 | 0.08 | 0.08 | 0.13 | 0.0004 |
| (J) 5xG3.5 (G3.5) | 0.18 | 0.07 | 0.00 | 0.00 | 0.00 | 0.03 | 0.25 | 0.25 | 0.0002 |

## A  Rule-based Parser

Below is a code snippet of all rules checked by the Rule-based parser. For full implementation refer to our GitHub[2].

```
SUBJECT = "<subject>"
OBJECT = "<object>"
errors = []

# SUBJECT entity errors:
if SUBJECT not in text:
    errors.append(TemplateErrors.NO_SUBJECT)
elif text.count(SUBJECT) > 1:
    errors.append(TemplateErrors.
        MULTIPLE_SUBJECTS)
# OBJECT entity errors:
if OBJECT not in text:
    errors.append(TemplateErrors.NO_OBJECT)
elif text.count(OBJECT) > 1:
    errors.append(TemplateErrors.MULTIPLE_OBJECTS
        )
# PLACEHOLDER mismatch errors
# e.g. `<author>` instead of `<subject>`/`<
    object>`
if contains_illegal_placeholder(text):
    errors.append(TemplateErrors.
        ILLEGAL_PLACEHOLDER)
```

## B  Tools and Repositories

We used Python 3.9 and 3.10 to run our experiments with LangChain[3] and OpenAI API[4] for efficient work with large language model pipelines. Metrics were calculated using the following existing implementations:

- BLEU/METEOR: GEM-metrics benchmark[5]

- PARENT: multiprocessing variant from Clément Rebuffel[6]

- BLEURT: code from google-research[7] with default BLEURT-20 checkpoint[8]

- NUBIA: original NUBIA repository[9]

---

[2] https://github.com/vejvarm/ASPIRO/blob/0f86ead6218b0100aff650656ef1ca9a8e2e485c/parsing.py

[3] python.langchain.com
[4] platform.openai.com/docs/api-reference
[5] github.com/GEM-benchmark/GEM-metrics
[6] github.com/KaijuML/parent
[7] github.com/google-research/bleurt
[8] bleurt/blob/master/checkpoints.md
[9] github.com/wl-research/nubia

## C  Metrics

For comparability of our automatic metric evaluations (§4.1), we leverage most of the lexical and semantic similarity metrics used by Kasner et al. (2023). Below is a brief explanation of their significance.

**BLEU** (Papineni et al., 2002) and **METEOR** (Banerjee and Lavie, 2005) are metrics that quantify the lexical similarity between model-generated outputs and (typically human-produced) references by utilising n-gram overlap.

**PARENT** (Dhingra et al., 2019) additionally assesses n-gram overlap of generated outputs with source structured data (i.e., table), which acts as an auxiliary reference beyond the reference text. This metric rewards instances where the hypothesis encapsulates all the information derived from the table, even if some elements are absent from the reference. Conversely, the PARENT score discriminates situations where the reference text contains supplementary information, which is absent in the structured data, implying the integration of external knowledge or embellishments in the reference.

**BLEURT** (Sellam et al., 2020) is a trained metric that complements the above lexical similarity metrics by capturing semantic similarity.

**NUBIA** (Kane et al., 2020) is also a trained metric that combines multiple sub-metrics to assess the interchangeability or equivalence of two texts. On the surface, this metric generates a single score (**NB**), that ranges between 0 and 1. Similar to Kasner et al. (2023), we also report on the sub-metrics which are used for the total NB value:

**SS**  Semantic Similarity operates on a scale of 0-5, where higher values suggest higher semantic similarity

**C%**  Contradiction percentage increases as the output and reference contradict each other in their meaning.

**N%**  Neutral percentage (also referred to as "chance of irrelevancy")[10] increases if the output contains new information or information which is irrelevant to the reference.

**E%**  Entailment percentage increases as the information in reference is entailed by the model output.

## D  Datasets

### D.1  DART

DART dataset introduced by Nan et al. (2021) is a large <triple-set, sentence> pair dataset aggregated from WikiSQL, WikiTableQuestions, WebNLG2017 and CleanedE2E with 62,659/6,980/12,552 samples in the train/dev/test sets respectively. All splits combined have total of 4299 unique relations. Each sample contains a set of up to 7 RDF triples with different relations, and human-crafted sentences as natural verbalisations of the given structured data.

### D.2  DART-SingleRDF Dataset

As DART (§D.1) combines multiple relations within one sample, we opted to extract only samples with single triple in the input set from all DART splits and combine them into DART-SingleRDF subset. DART-SingleRDF contains a total of 2,947 <single-triple, sentence> entries with 1,439 unique relations.

### D.3  Rel2Text

Rel2Text is specifically designed by Kasner et al. (2023) for the verbalisation task of single-triples focused on out-of-domain and previously unseen relations. It contains a total of 4,097 <relation, description, single-triple, verbalisation> samples. In §4.1, we use the test split of Rel2Text[11] with 616 total samples and 226 unique relations, unseen in the training set.

### D.4  WebNLG

WebNLG is commonly used dataset by Gardent et al. (2017) where entries contain up to 7 triples extracted from DBpedia categories and labelled by human-crafted sentences. We specifically use the enrichment of WebNLG version 1.4 from (Castro Ferreira et al., 2018), which contains 354 unique relations in total.

## E  Additional Experiments

### E.1  Automatic metrics for DART-SingleRDF

We used DART-SingleRDF (§D.2) as test set for automatic metric evaluation of ASPIRO. The **Full** results are reported in Table 8 and **Reduced** results in Table 9.

---

[10]github.com/wl-research/nubia

[11]github.com/kasnerz/rel2text/tree/main/data/full

Table 8: Evaluation of ASPIRO with $\mathcal{T}_I$ =(**A**)SDOT/(**J**)SON and $\mathcal{L}_0$ =G3.5/G3.5T on DART-SingleRDF dataset using automatic metrics. See Table 2 for for model descriptions, C for metric descriptions.

| all samples (2947) | bleu ↑ | meteor ↑ | BLEURT ↑ | NB ↑ | SS ↑ | C (%) ↓ | N (%) ↓ | E (%) ↑ | PARENT$_{F1}$ ↑ |
|---|---|---|---|---|---|---|---|---|---|
| (**A**) G3.5≡ $\mathcal{L}_0$ | 34.41 | 34.48 | 0.69 | 0.67 | 4.00 | 16.97 | **14.50** | 68.53 | 0.8257 |
| (**A**) $\mathcal{L}_0$+1xG3.5T | 33.60 | 34.19 | 0.68 | 0.67 | 4.01 | 15.56 | 15.20 | 69.24 | 0.8286 |
| (**A**) $\mathcal{L}_0$+1xG3.5 | 34.11 | 34.38 | 0.69 | 0.67 | 4.00 | 17.04 | 14.50 | 68.47 | 0.8250 |
| (**A**) $\mathcal{L}_0$+1xG4 | 33.23 | 34.39 | 0.69 | 0.67 | 4.01 | 16.82 | 14.60 | 68.58 | 0.8261 |
| (**A**) $\mathcal{L}_0$+5xG3.5T | **33.69** | **34.59** | 0.69 | **0.68** | **4.07** | **14.02** | 15.36 | **70.61** | **0.8348** |

| all samples (2947) | bleu ↑ | meteor ↑ | BLEURT ↑ | NB ↑ | SS ↑ | C (%) ↓ | N (%) ↓ | E (%) ↑ | PARENT$_{F1}$ ↑ |
|---|---|---|---|---|---|---|---|---|---|
| (**A**) G3.5T≡ $\mathcal{L}_0$ | 33.76 | 35.25 | 0.71 | 0.71 | 4.12 | 14.53 | **14.32** | 71.16 | 0.8290 |
| (**A**) $\mathcal{L}_0$+1xG3.5T | 34.23 | 35.58 | 0.71 | 0.72 | 4.19 | 11.38 | 14.49 | 74.13 | 0.8347 |
| (**A**) $\mathcal{L}_0$+1xG3.5 | 34.03 | 35.88 | 0.71 | 0.73 | 4.20 | 11.16 | 14.52 | 74.32 | 0.8342 |
| (**A**) $\mathcal{L}_0$+1xG4 | 33.48 | 35.67 | 0.71 | 0.73 | 4.20 | 11.29 | 14.75 | 73.96 | 0.8341 |
| (**A**) $\mathcal{L}_0$+5xG3.5T | **34.25** | **35.91** | 0.71 | **0.73** | **4.24** | **9.84** | 14.97 | **75.18** | **0.8406** |

| all samples (2947) | bleu ↑ | meteor ↑ | BLEURT ↑ | NB ↑ | SS ↑ | C (%) ↓ | N (%) ↓ | E (%) ↑ | PARENT$_{F1}$ ↑ |
|---|---|---|---|---|---|---|---|---|---|
| (**J**) G3.5≡ $\mathcal{L}_0$ | **35.49** | 36.96 | **0.72** | **0.76** | 4.42 | 4.68 | 12.05 | 83.27 | 0.8534 |
| (**J**) $\mathcal{L}_0$+1xG3.5T | 35.16 | 36.64 | 0.72 | 0.75 | 4.40 | 4.86 | 12.13 | 83.01 | 0.8517 |
| (**J**) $\mathcal{L}_0$+1xG3.5 | 35.48 | 36.95 | 0.72 | 0.76 | 4.42 | 4.67 | 11.98 | 83.35 | 0.8534 |
| (**J**) $\mathcal{L}_0$+1xG4 | 35.49 | **36.97** | 0.72 | 0.76 | **4.43** | **4.62** | 11.95 | **83.43** | **0.8535** |
| (**J**) $\mathcal{L}_0$+5xG3.5T | 35.44 | 36.90 | 0.72 | 0.76 | 4.42 | 4.66 | **11.94** | 83.41 | 0.8532 |

| all samples (2947) | bleu ↑ | meteor ↑ | BLEURT ↑ | NB ↑ | SS ↑ | C (%) ↓ | N (%) ↓ | E (%) ↑ | PARENT$_{F1}$ ↑ |
|---|---|---|---|---|---|---|---|---|---|
| (**J**) G3.5T≡ $\mathcal{L}_0$ | 32.76 | 36.34 | **0.70** | **0.74** | 4.36 | 5.56 | 16.03 | 78.41 | **0.8408** |
| (**J**) $\mathcal{L}_0$+1xG3.5T | 31.13 | 35.82 | 0.70 | 0.73 | 4.33 | 5.72 | 15.91 | 78.38 | 0.8375 |
| (**J**) $\mathcal{L}_0$+1xG3.5 | 32.76 | 36.35 | 0.70 | 0.74 | 4.36 | 5.55 | 15.94 | 78.51 | 0.8408 |
| (**J**) $\mathcal{L}_0$+1xG4 | **32.89** | **36.40** | 0.70 | 0.74 | **4.37** | **5.47** | **15.67** | **78.87** | 0.8408 |
| (**J**) $\mathcal{L}_0$+5xG3.5T | 32.13 | 36.15 | 0.70 | 0.73 | 4.35 | 5.63 | 16.01 | 78.36 | 0.8400 |

Table 9: (Subsets of only generated templates with $F1(\mathcal{L}_0) < 0.7$) Evaluation of ASPIRO with $\mathcal{T}_I$ =(**A**)SDOT/(**J**)SON and $\mathcal{L}_0$ =G3.5/G3.5T on DART-SingleRDF dataset using automatic metrics. See Table 2 for for model descriptions, C for metric descriptions.

| $F1(\mathcal{L}_0) < 0.7$ only (**491**) | bleu ↑ | meteor ↑ | BLEURT ↑ | NB ↑ | SS ↑ | C (%) ↓ | N (%) ↓ | E (%) ↑ | PARENT$_{F1}$ ↑ |
|---|---|---|---|---|---|---|---|---|---|
| (**A**) G3.5≡ $\mathcal{L}_0$ | 16.76 | 24.20 | 0.57 | 0.49 | 3.19 | 27.74 | 23.56 | 48.70 | 0.5324 |
| (**A**) $\mathcal{L}_0$+1xG3.5T | 17.09 | 24.59 | 0.56 | 0.49 | 3.27 | 24.12 | 24.60 | 51.27 | 0.5470 |
| (**A**) $\mathcal{L}_0$+1xG3.5 | 16.09 | 24.01 | 0.56 | 0.48 | 3.17 | 28.09 | 23.37 | 48.55 | 0.5331 |
| (**A**) $\mathcal{L}_0$+1xG4 | 15.66 | 24.49 | 0.57 | 0.49 | 3.25 | 27.43 | **22.47** | 50.10 | 0.5397 |
| (**A**) $\mathcal{L}_0$+5xG3.5T | **18.10** | **25.76** | **0.58** | **0.52** | **3.46** | **20.70** | 23.60 | **55.69** | **0.5706** |

| $F1(\mathcal{L}_0) < 0.7$ only (**473**) | bleu ↑ | meteor ↑ | BLEURT ↑ | NB ↑ | SS ↑ | C (%) ↓ | N (%) ↓ | E (%) ↑ | PARENT$_{F1}$ ↑ |
|---|---|---|---|---|---|---|---|---|---|
| (**A**) G3.5T≡ $\mathcal{L}_0$ | 17.24 | 25.13 | 0.60 | 0.55 | 3.45 | 23.09 | **22.08** | 54.83 | 0.5300 |
| (**A**) $\mathcal{L}_0$+1xG3.5T | 18.16 | 26.12 | 0.61 | 0.58 | 3.58 | 17.24 | 22.56 | 60.21 | 0.5513 |
| (**A**) $\mathcal{L}_0$+1xG3.5 | 17.71 | 26.70 | **0.62** | 0.58 | 3.60 | 18.28 | 22.90 | 58.81 | 0.5453 |
| (**A**) $\mathcal{L}_0$+1xG4 | 16.74 | 26.31 | 0.62 | 0.58 | 3.61 | 18.29 | 22.90 | 58.82 | 0.5499 |
| (**A**) $\mathcal{L}_0$+5xG3.5T | **19.42** | **27.24** | 0.62 | **0.59** | **3.73** | **14.16** | 23.22 | **62.63** | **0.5691** |

| $F1(\mathcal{L}_0) < 0.7$ only (**334**) | bleu ↑ | meteor ↑ | BLEURT ↑ | NB ↑ | SS ↑ | C (%) ↓ | N (%) ↓ | E (%) ↑ | PARENT$_{F1}$ ↑ |
|---|---|---|---|---|---|---|---|---|---|
| (**J**) G3.5≡ $\mathcal{L}_0$ | 19.23 | 28.26 | **0.64** | **0.64** | 4.01 | 7.95 | 17.79 | 74.26 | 0.5317 |
| (**J**) $\mathcal{L}_0$+1xG3.5T | 19.34 | 28.19 | 0.63 | 0.64 | 4.00 | 7.99 | 18.14 | 73.87 | 0.5310 |
| (**J**) $\mathcal{L}_0$+1xG3.5 | 19.22 | 28.21 | 0.64 | 0.64 | 4.01 | 8.03 | 17.89 | 74.08 | 0.5316 |
| (**J**) $\mathcal{L}_0$+1xG4 | 19.27 | **28.34** | 0.64 | 0.64 | **4.02** | **7.89** | **17.62** | **74.49** | **0.5322** |
| (**J**) $\mathcal{L}_0$+5xG3.5T | **19.42** | 28.08 | 0.63 | 0.64 | 4.00 | 7.91 | 17.80 | 74.29 | 0.5304 |

| $F1(\mathcal{L}_0) < 0.7$ only (**387**) | bleu ↑ | meteor ↑ | BLEURT ↑ | NB ↑ | SS ↑ | C (%) ↓ | N (%) ↓ | E (%) ↑ | PARENT$_{F1}$ ↑ |
|---|---|---|---|---|---|---|---|---|---|
| (**J**) G3.5T≡ $\mathcal{L}_0$ | 16.70 | 27.25 | 0.61 | 0.61 | 3.91 | 8.45 | 21.34 | 70.21 | 0.5313 |
| (**J**) $\mathcal{L}_0$+1xG3.5T | 14.31 | 26.01 | 0.60 | 0.60 | 3.82 | 8.56 | 21.81 | 69.63 | 0.5215 |
| (**J**) $\mathcal{L}_0$+1xG3.5 | 16.44 | 27.27 | 0.61 | 0.61 | 3.91 | 8.42 | **21.23** | 70.35 | 0.5314 |
| (**J**) $\mathcal{L}_0$+1xG4 | **16.73** | **27.43** | **0.62** | **0.62** | **3.93** | **8.08** | 21.31 | **70.61** | **0.5325** |
| (**J**) $\mathcal{L}_0$+5xG3.5T | 16.04 | 26.75 | 0.61 | 0.60 | 3.87 | 8.35 | 22.18 | 69.47 | 0.5296 |

**Full:** Results in Table 8 show slight or no discrepancies in the metrics between all the experiments, which could be attributed to variational error. Considering the high API model costs (§F), we did not run DART experiments multiple times to provide deviations. Instead, we **reduce** the data to only *problematic* samples by taking a subset of $y_0^r = \mathcal{L}_0(x^r)$ generated templates which satisfy $\mathsf{PARENT}_{F1}(y_0^r) < 0.7$. In other words, we take a subset of samples, for which outputs of 0-shot model in the respective sub-table were flagged as inconsistent by the Consistency Validator (§3.2) using $\mu = 0.7$. We report the same metric evaluation process in Table 9.

**Discussion:** For the **Reduced** evaluation in Table 9, we found that ASPIRO shows significant improvement only when **(A)SDOT** is initial prompt and only with 5-shot gpt-3.5-turbo setting. A point of interest is also the Neutral (irrelevance) score (**N %**), which the 5-shot setting generally increases, suggesting the $N$-shot setting is reducing the relevance of generated verbalisations to the references. For JSON prompts 1-shot gpt-4 setting has slight, albeit marginal lead over other settings.

### E.2 Performance on WebNLG

We additionally evaluated performance on WebNLG (§D.4), following a similar approach as with Rel2Text in the main experiments (§4). GPT-3.5-turbo-0301 is used as LLM instances of all calls to both N-shot generator LLM stack and Consistency Validator.

**Parsing errors:** We observed (Table 10) that for WebNLG, ASPIRO is generally not able to fix any errors and CV conversely increases the number of total errors, making 3 of the templates more "flawed" than without CV.

**Automatic metrics:** We compare the templates generated by ASPIRO to the manually crafted templates from Kasner and Dusek (2022) to evaluate lexical similarity using PARENT, BLEU and METEOR. The results, seen in Table 11, are marginal at best and we can only observe improvement in BLEU score, while PARENT F1 and METEOR are highest for zero-shot setting. Due to time restraints, we did not include BLEURT and NUBIA evaluations.

**Conclusion:** Contrary to our original belief, we can conclude that ASPIRO pipeline does not provide significant improvement over 0-shot method on the WebNLG dataset.

## F Run time and cost of ASPIRO

\$191 US is the overall expenditure for OpenAI API calls during our experiments. However, it is important to note that we made many redundant API calls in the development of ASPIRO so the necessary costs should be lower. The main bulk of the costs amounts to GPT3-davinci and text-davinci-003 calls.

### F.1 Run time analysis

Table 12 presents the average run time in seconds across five experimental runs using the WebNLG dataset, which comprises 354 unique relations. This translates to a total of 354 calls required for the 0x model, a zero-shot call to the initial model (GPT3.5-turbo). Subsequent retry shots only need as many calls as there are templates with parsing errors.

**Cumulative mean time:** Given the nature of our experiments, where subsequent runs leverage results from preceding runs (for instance, the 2-shot run utilises results from the 1-shot run and only re-prompts those with parsing errors), we introduce *Cumulative mean time* to illustrate the total time necessary to execute all shots of the respective experiment.

### F.2 Estimated API call costs

For a "worst-case-scenario" cost estimate of AS-PIRO (all templates are tagged for retry shot), we made calculations for the GPT3.5-turbo model, which charges \$0.002 per 1000 tokens (as of the time of our experiments). Table 13 provides cost estimations for the experiments conducted on the Rel2Text, WebNLG, and DART datasets using GPT3.5-turbo. To derive the costs associated with the GPT3-davinci or text-davinci-003 models (charged at \$0.02 per 1000 tokens), multiply the presented Table 13 values by a factor of 10.

Table 10: Number of errors tagged in generated templates by our Rule-based parser (§3.1.2) for different experiment setups with gpt3.5-turbo-0301 on the full Enriched WebNLG v1.4 dataset. All models (base, retry and CV) are instances of GPT-3.5-turbo-0301. Multiple SUBJECTs column was ommited (0 for all experiments).

| WebNLG dataset 354 unique templates | Total Errors | Template Errors | Missing SUBJECT | Missing OBJECT | Multiple OBJECTs | Illegal PLACEHOLDER |
|---|---|---|---|---|---|---|
| **(0shot) ()** | 80 | 79 | 24 | 56 | 0 | 0 |
| **(0shot) (CV)** | 83 | 79 | 23 | 57 | 2 | 1 |
| **(1shot) ()** | 80 | 79 | 22 | 58 | 0 | 0 |
| **(2shot) ()** | 81 | 80 | 23 | 58 | 0 | 0 |
| **(3shot) ()** | 80 | 79 | 23 | 57 | 0 | 0 |
| **(4shot) ()** | 80 | 79 | 22 | 58 | 0 | 0 |
| **(5shot) ()** | 80 | 79 | 22 | 58 | 0 | 0 |
| **(5shot) (CV)** | 83 | 79 | 23 | 57 | 2 | 1 |

Table 11: Automatic Metrics on WebNLG v1.4 against human-crafted verbalisation templates from Kasner and Dusek (2022).

| 0shot model = G3.5T (retry model) (Nshot) (CV model) | $\text{PARENT}_P$ | $\text{PARENT}_R$ | $\text{PARENT}_{F1}$ | BLEU | METEOR (%) |
|---|---|---|---|---|---|
| **() (0shot) ()** | 0.7455 | 0.9594 | 0.8308 | 54.50 | 46.71 |
| **() (0shot) (G3.5T)** | 0.7461 | 0.9524 | **0.8278** | 55.00 | **46.20** |
| **(G3.5T) (1shot) ()** | 0.7454 | 0.9591 | 0.8305 | 54.37 | 46.54 |
| **(G3.5T) (2shot) ()** | 0.7455 | 0.9592 | 0.8306 | 54.44 | 46.56 |
| **(G3.5T) (3shot) ()** | 0.7456 | 0.9595 | 0.8308 | 54.47 | 46.58 |
| **(G3.5T) (4shot) ()** | 0.7456 | 0.9595 | 0.8308 | 54.44 | 46.59 |
| **(G3.5T) (5shot) ()** | 0.7456 | 0.9595 | 0.8308 | 54.44 | 46.59 |
| **() (5shot) (G3.5T)** | 0.7461 | 0.9537 | **0.8285** | 54.89 | **46.34** |

Table 12: Average run time (in seconds) of experimental runs on the WebNLG dataset with the GPT3.5-turbo model.

| Experiment | Mean time (s) | Cumulative mean time (s) |
|---|---|---|
| **0x** | 377.5 | 377.5 |
| **0x (CV)** | 173.3 | 550.8 |
| **1x** | 142.4 | 519.9 |
| **2x** | 118.3 | 638.1 |
| **3x** | 159.3 | 797.5 |
| **4x** | 111.7 | 909.2 |
| **5x** | 108.1 | 1017.2 |
| **5x (CV)** | 138.8 | 1156.0 |

Table 13: Calculated API call cost (USD) estimations for the experiments conducted on the Rel2Text, WebNLG, and DART datasets using the GPT3.5-turbo (x10 for GPT3-davinci and text-davinci-003) model with either ASDOT initial prompt (A) or JSON initial prompt (J). The costs are detailed for both individual N-Shot settings and Consistency Validation, with the total cost calculated for each experiment setting.

| ($\mathcal{T}_I$) dataset NxModel | Total N-Shot | Total CV (G3.5T) | Total Cost (G3.5T) |
|---|---|---|---|
| **(A) Rel2Text 0xG3.5T** | 0.07 | 0.14 | 0.22 |
| **(A) Rel2Text 1xG3.5T** | 0.14 | 0.14 | 0.29 |
| **(A) Rel2Text 5xG3.5T** | 0.43 | 0.14 | 0.58 |
| **(J) Rel2Text 0xG3.5T** | 0.14 | 0.14 | 0.28 |
| **(J) Rel2Text 1xG3.5T** | 0.27 | 0.14 | 0.42 |
| **(J) Rel2Text 5xG3.5T** | 0.81 | 0.14 | 0.96 |
| **(A) WebNLG 0xG3.5T** | 0.11 | 0.23 | 0.34 |
| **(A) WebNLG 1xG3.5T** | 0.23 | 0.23 | 0.45 |
| **(A) WebNLG 5xG3.5T** | 0.68 | 0.23 | 0.91 |
| **(J) WebNLG 0xG3.5T** | 0.21 | 0.23 | 0.44 |
| **(J) WebNLG 1xG3.5T** | 0.42 | 0.23 | 0.65 |
| **(J) WebNLG 5xG3.5T** | 1.27 | 0.23 | 1.50 |
| **(A) DART 0xG3.5T** | 0.46 | 0.92 | 1.38 |
| **(A) DART 1xG3.5T** | 0.92 | 0.92 | 1.84 |
| **(A) DART 5xG3.5T** | 2.76 | 0.92 | 3.68 |
| **(J) DART 0xG3.5T** | 0.86 | 0.92 | 1.78 |
| **(J) DART 1xG3.5T** | 1.73 | 0.92 | 2.65 |
| **(J) DART 5xG3.5T** | 5.18 | 0.92 | 6.10 |

## G  Prompt Templates

### G.1  Initial Prompt: ASDOT

Table: Michael | birth Place | USA
Text: Michael was born in the USA.

Table: First Clearing | location | On NYS 52 1 Mi.
  Youngsville
Text: First Clearing is located at On NYS 52 1 Mi.
  Youngsville.

Table: Abilene Regional Airport | city Served |
  Abilene Texas
Text: Abilene Regional Airport serves Abilene Texas.

Table: Alfred Moore Scales | active Years Start Date |
  1875-03-04
Text: Alfred Moore Scales started to be active on
  1875-03-04.

{example_table_str}
Text:

### G.2  Initial Prompt: JSON

### example:
``` json
{{"intput_data": [["World Trade Center", "architect",
  "Minoru Yamasaki"], ["Seymour Centre", "architect
  ", "Allen Jack+Cottier"]]}}
```

``` json
{{"subject_entities": ["World Trade Center", "Seymour
  Centre"],
"relation": "architect",
"object_entities": ["Minoru Yamasaki", "Allen Jack+
  Cottier"],
"agnostic_template": "<object> is the architect of <
  subject>."
}}
```
###

### your task:
``` json
{{"input_data": {example_rdf_list}}}
```

``` json
{{

### G.3  Retry Prompt[12]

```` Completion
{completion}
````

Above, the Completion did not satisfy the constraints
  given in the Prompt.
Details: `{error}`
Please try again.

```` Prompt
{prompt}
````

### G.4  Consistency Prompt

Your task is to evaluate a [string] based on the
  following [rules] and output a `valid` flag
  which is either 1 ([string] complies with [rules
  ]) or 0 ([string] breaks some [rules]) and an `
  advice` which explains in one short sentence how
  to fix [string] to comply with all [rules] (if
  valid==1, advice=""). You will also output a `
  valid_string`, which will follow the advice and
  comply to the rules (if valid==1, valid_string=[
  string])
[rules]:
– [string] must contain exactly one `<subject>`
  substring.
– [string] must contain exactly one `<object>`
  substring.
– [string] must **not contain any named entities or
  specific references other than `<subject>` and
  `<object>`**.
– [string] should align with the semantic meaning of
  the [relation] provided, without adding or
  implying any information beyond it.
– [string] must concisely and factually represent the
  information in [relation], without embellishment
  or unnecessary details.
[string]: {template}
[relation]: {data}
``` json
{{
    "valid":
    "advice":
    "valid_string":
}}
```