# OpenReview forum: "ASPIRO: Any-shot Structured Parsing-error-Induced ReprOmpting for Consistent Data-to-Text Generation"
_EMNLP/2023/Conference — EMNLP 2023 Findings_

### Official Review · Reviewer_eDmS · 2023-07-28

**Soundness:** 3

**Excitement:**

3: Ambivalent: It has merits (e.g., it reports state-of-the-art results, the idea is nice), but there are key weaknesses (e.g., it describes incremental work), and it can significantly benefit from another round of revision. However, I won't object to accepting it if my co-reviewers champion it.

**Paper Topic And Main Contributions:**

The paper proposes a method for verbalizing single RDF triples - a subtask of data-to-text generation. Unlike previous works which rely on templates or finetuned language models, the proposed method - ASPIRO - utilizes a processing pipeline based on prompting large language models (LLMs).

The pipeline consists of two components. In the first component, a LLM is prompted for producing a verbalization of the triple and a simple rule-based parser is used to either accept the verbalization or iteratively re-prompt a LLM given the original input and the parsing errors. The second component further prompts a LLM to improve the consistency of the generated verbalization.

The results show that the proposed LLM-based pipeline achieves competitive results on verbalizing single triple to the finetuned BART model proposed in Kasner et al. (2023).

**Reasons To Accept:**

- The work shows that LLMs can be used for generating single RDF triples, reducing the need for training data for this task.
- The work introduces multiple components for iterative validation of the model outputs, improving the controllability of the generation process.

**Reasons To Reject:**

- The paper does not read well; the text is cluttered with math formulations which obscure the main points of the paper.
- The processing pipeline seems too heavy for the proposed task. Crucially, no ablation study is provided to assess the importance of individual components compared e.g. to simply prompting a single LLM with variety of prompts.
- Although the authors provide code / algorithm snippets throughout the paper, there is no mention of releasing their code upon publication which may hinder reproducibility.
- Overall, would rather suggest transforming this paper into a long paper and presenting more principled experiments: comparing the proposed architecture with simpler baselines and performing and ablation study on indvidual components.

**Reproducibility:**

3: Could reproduce the results with some difficulty. The settings of parameters are underspecified or subjectively determined; the training/evaluation data are not widely available.

**Reviewer Confidence:**

4: Quite sure. I tried to check the important points carefully. It's unlikely, though conceivable, that I missed something that should affect my ratings.

**Typos Grammar Style And Presentation Improvements:**

- 189 - BERT -> BART
- BLEU and METEOR scores are usually capitalized

---

> ### Author Rebuttal · Authors · 2023-08-28
>
> Thank you for your thorough review and the comments and suggestions you provided. We would like to answer some of your concerns about this study below. Please also feel free to read the rebuttals for the other reviews on this paper for additional data and information.
>
> ## ad. Reasons to Reject:
> ### 1) Paper readability
> ("*The paper does not read well; the text is cluttered with math formulations which obscure the main points of the paper.*")
> #### Response
> We have rewritten the Methods section in more complete and granular way, as it was indeed very cluttered. Please see it at the end of this rebuttal (formatting is off, as it is converted from latex to markdown).
> ### 2) Ablation study
> ("*The processing pipeline seems too heavy for the proposed task. Crucially, **no ablation study** is provided to assess the importance of individual components compared e.g. to simply prompting a single LLM with variety of prompts.*")
> #### Response
> As we mention in the **Operational costs** paragraph of the **Limitations** section, we believe that the consistency validator is not significantly beneficial to ASPIRO performance. We have conducted an ablation study on Rel2Text dataset, but due to the monthly cost hard-cap on OpenAI API it was not very thorough and we did not give it enough focus. Below is the result of our experiments. First 2 rows present GPT3.5Turbo as the base model for all parts of ASPIRO, while rows 3 and 4 use GPT3-davinci as the base model.
>
> Number of parsing error counts without and with (**CV** ) consistency validation:
>
> | **error counts**                     | **Total Errors** | **Template Errors** | **Missing SUBJECT** | **Missing OBJECT** | **Illegal PLACEHOLDER** |
> |--------------------------------------|:----------------:|:-------------------:|:-------------------:|:------------------:|:-----------------------:|
> | **5xGPT3.5Turbo**                    |        12        |          11         |          2          |         10         |            0            |
> | **5xGPT3.5Turbo (CV GPT3.5Turbo)**   |        10        |          9          |          2          |          8         |            0            |
> | **5xGPT3-davinci**                   |        23        |          21         |          5          |         18         |            0            |
> | **5xGPT3-davinci (CV GPT3-davinci)** |        16        |          14         |          3          |         13         |            0            |
>
> Automatic metrics without and with (**CV** ) consistency validation:
>
> | **automatic metrics for Rel2Text test set** | **BLEU** | **METEOR (%)** | **BLEURT** | **NB** | **SS** | **C (%)** | **I (%)** | **E (%)** | **PARENT_F1** |
> |---------------------------------------------|:--------:|:--------------:|:----------:|:------:|:------:|:---------:|:---------:|:---------:|:-------------:|
> | **5xGPT3.5Turbo**                           |   50.88  |      44.97     |   0.8215   | 0.8725 |  4.73  |    3.36   |    8.04   |   88.59   |     0.8910    |
> | **5xGPT3.5Turbo (CV GPT3.5Turbo)**          |   50.89  |      44.95     |   0.8213   | 0.8718 |  4.73  |    3.02   |    8.34   |   88.64   |     0.8908    |
> | **5xGPT3-davinci**                          |   50.96  |      44.69     |   0.8200   | 0.8680 |  4.71  |    4.92   |    7.59   |   87.48   |     0.8879    |
> | **5xGPT3-davinci (CV GPT3-davinci)**        |   51.32  |      44.77     |   0.8200   | 0.8702 |  4.72  |    3.35   |    7.70   |   88.94   |     0.8895    |
>
> The above tables show, that CV is effective at reducing the Parsing error count slightly (12 to 10 and 23 to 16), but only shows marginal gains on statistical metrics.
> We will additionally include these ablation results and the discussion on the importance of Consistency Validation in the appendix.
> ### 3) Code and reproducibility
> ("*Although the authors provide code / algorithm snippets throughout the paper, there is no mention of releasing their code upon publication which may hinder reproducibility.*")
> #### Response:
> - We did provide both working code as well as all result data in the supplementary files during submission.
> - We did not include GitHub mention as we feared it would clash with the anonymity requirements. We do have working code prepared for GitHub and we will add a reference to it in the final paper version.
> ### 4) Paper format
> ("*Overall, would rather suggest transforming this paper into a long paper and presenting more principled experiments: comparing the proposed architecture with simpler baselines and performing and ablation study on indvidual components.*")
> #### Response:
> This is a valid point. We had problems with including all the results into 4 pages of content. We are discussing with the PCs the possibility of transforming the paper from short to long format in the final submission.
>
> # Methods (revised)
> The proposed method (ASPIRO) revolves around the conversion of structured data samples into verbalisation templates using a two-stage pipeline: **$N$-shot Generator** (§subsec:n-shot-generator) and **Consistency Validator** (§subsec:consistency-validator). The pipeline processes structured data samples, wherein each sample comprises of one or more RDF triples which share the same relation. ASPIRO (see Figure `fig:ASPIRO-schema`) starts with an initial prompt to verbally articulate the structured data. This is equivalent to prompting a single LLM directly. If the zeroth attempt isn't accurate, it will retry a maximum of $N$ times, refining the previous completion based on parsing errors (§subsec:rule-based-parser). Subsequently, the outputs are validated for consistency, ensuring faithful and reliable verbalisations. We explain the individual stages and their sub-modules in the sections below. Refer to Figure `fig:ASPIRO-schema` for full pipeline and terminology and Figure `fig:ASPIRO-example` for a more illustrative example of ASPIRO pipeline on a single input.
>
> ## $N$-shot Generator
> §subsec:n-shot-generator
>
> $N$-shot Generator further fractures into an LLM stack and a Rule-based parser. The LLM Stack is tasked with generating verbalisation attempts based on given initial prompt (see §subsec:app-prompt-asdot). It does so with the help of the Rule-based parser. This parser checks the generated completions for structural accuracy, ensuring they adhere to expected patterns.
>
> ### LLM Stack
> §subsec:llm-stack
>
> The LLM stack is a set of $N+1$ LLMs, indexed from $0$ to $N$, where $\mathcal{L}_0$ is responsible for the initial completion, whereas each further retry shot initiated by the Rule-based parser (§subsec:rule-based-parser) increments the index by $1$. Each $L_n$ is instantiated separately and does not have to be the same model. Equation (1) shows the single completion for structured input sample $x$ at shot $n`.
>
> $$
> y_n = \mathcal{L}_n(\mathcal{T}(x)) \tag{1}
> $$
>
> where $\mathcal{T}$ is a given prompt and can be either $\mathcal{T}_I$ (initial) or $\mathcal{T}_R$ (retry).
>
> ### Rule-based parser
> §subsec:rule-based-parser
>
> A purely algorithmic module, which validates $y_n$ against a set of conditions $\{\mathcal{C}\}$ one by one. If $y_n$ does not pass the condition $C_i$, a respective parsing error is logged into set $\mathcal{E}_n$. The rules for each given completion are as follows:
>
> $\mathcal{C}_0$ ... has exactly one `<subject>` substring.
>
> $\mathcal{C}_1$ ... has exactly one `<object>` substring.
>
> $\mathcal{C}_2$ ... has no other `<...>` substrings.
>
> If the parser identifies any error in the structure, the next LLM in the LLM stack is re-prompted with Retry Prompt (§subsec:app-prompt-retry) to generate new completion.
>
> ## Consistency Validator
> §subsec:consistency-validator
>
> Even if the outputs from the $N$-shot Generator adhere to the structural patterns, they might still contain inaccuracies, such as hallucinated content. This module assesses the quality of the verbalisations, using the PARENT statistical metric [dhingra-etal-2019-handling]. If PARENT F1 score is too low, the module will utilise an LLM with specialised Consistency Prompt (§subsec:app-prompt-consistency) to improve the sentence.
>
> ### PARENT$_{F1}$ threshold
> §subsec:parent-score
>
> To gauge the quality of the completion $y_n$ from $N$-shot Generator, we set a minimal threshold ($\mu$) for the PARENT score of $y_n$. The score is calculated using eq. (3) against artificially constructed table and reference.
>
> First, we construct the respective hypothesis, table and reference entries:
>
> $$ h = y_n\texttt{.replace}([s, o], e) \\ t = \langle e, r\texttt{.split(" ")}, e \rangle \\ \rho = r \tag{2}
> $$
>
> where `<subject>` and `<object>` are replaced with `<entity>` to prevent penalizing order discrepancy between hypothesis and table.
>
> We then calculate the PARENT F1 score using equation (3).
>
> $$
> F1(y_n) = \texttt{PARENT}(h, \rho, t) \tag{3}
> $$
>
> ### Consistency LLM
> §subsec:consistency-llm
>
> If the calculated PARENT score from §subsec:parent-score is not sufficient, we call another LLM with prompt $\mathcal{T}_C$ as in eq. (4).
>
> $$
> y_C = \mathcal{L}_C(\mathcal{T}_C(r, y_n)) \tag{4}
> $$
>
> The prompt $\mathcal{T}_C$ is designed to guide $\mathcal{L}_C$ to identify problems with the given completion, provide advice how to fix it and subsequently produce fixed completion in a structured json output. See §subsec:app-prompt-consistency for full version of the prompt.
>
> ## Detailed Pipeline Formulation
> §subsec:formulation
>
> Given a dataset of structured data samples $\{x^r\}_{r\in\mathcal{R}}$, where $x^r=\{x^r_1, x^r_2, ..., x^r_m\}$ and $x^r_j$ is a single RDF triple $x^r_j=\langle s^r_j, r, o^r_j\rangle$ with relation $r\in \mathcal{R}$, the pipeline for one $x^r$ is as follows:
>
> **Step 0** Set $n=0$ and $\mathcal{T}^r_0=\mathcal{T}_I(x^r)$.
>
> **Step 1** Calculate $y^r_n$ using eq. (1).
>
> **Step 2** Use §subsec:rule-based-parser to validate $y^r_n$ against all conditions ${\mathcal{C}}$. If errors ($\mathcal{E}^r_n$)  are found, run equation (5) and return to **Step 1**. Otherwise go to **Step 3**.
>
> $$
>     \mathcal{T}^r_{n+1} = \mathcal{T}_R(x^r, y^r_n, \mathcal{E}^r_n) \\
>     n = n+1 \tag{5}
> $$
>
> **Step 3** Use §subsec:parent-score and calculate $F1(y^r_n)$ via eq. (3). If the calculated $F1$ score is lower than our chosen threshold $0\leq\mu \leq 1$, continue to **Step 4**. Otherwise, output current $y_n^r$ as the final completion $y^r$.
>
> **Step 4** Use §subsec:consistency-llm with prompt $\mathcal{T}_C^r$ and calculate $y^r_C$ via eq. (4).
>
> **Step 5** Compute \(F1\) scores of \(y_n^r\) and \(y_C^r\) using eq. (3) and take the completion with higher score via eq. (6) to produce the final completion
> $$
>     y^r = \underset{y\in\{y_n^r, y_C^r\}}{\texttt{argmax}(F1(y))} \tag{6}
> $$

---

### Official Review · Reviewer_ShqH · 2023-08-05

**Typos Grammar Style And Presentation Improvements:** Your title is misspelled. Consistent,…
**Soundness:** 3

**Excitement:**

3: Ambivalent: It has merits (e.g., it reports state-of-the-art results, the idea is nice), but there are key weaknesses (e.g., it describes incremental work), and it can significantly benefit from another round of revision. However, I won't object to accepting it if my co-reviewers champion it.

**Paper Topic And Main Contributions:**

This paper reports on an improvement to the method of Xiang et al. (2022) for generating template sentences from RDF triples (or other similar structured formats) using LLMs. The key improvement of this paper is the design of their pipeline which incorporates several methods of validating the LLM's output - first, they generate a sentence using an LLM prompt with several examples and a target RDF triple. They then run this through a rule-based parser to check the validity of the output relative to the schema. If it is not valid, they then reprompt the LLM with an updated prompt up to N times (5 max in their experiments) to try to get the LLM to output a valid sentence. Once the output passes the rule-based parser, they pass it to an LLM-based consistency validator for further validation and possible correction.

This setup results in significantly reduced erroneous outputs, as shown by their experiments on DART and Rel2Text.

**Reasons To Accept:**

This setup is a useful tool for generating text from structured data in a low-resource setting.

The paper is well written and the pipeline is clearly described.

The experimental results show the utility of the setup and its ability to reduce erroneous output versus a simpler single-prompting approach.

**Reasons To Reject:**

The complexity of the setup and the number of potential LLM calls leads me to believe it would be quite costly (as the authors admit in the limitations section). But they don't report on the actual cost. That would help bring the value of the proposed system into better focus. Additionally, run time would be useful to see, as this process seems like it could be quite time consuming.

The work largely build on previous work with an added method of repeating the prompting process for validation. That is important and valuable to show that this type of re-prompting helps, but it is relatively incremental.

I would like to have seen experiments with open source models; given the number of prompting calls necessary, it would be valuable to know if this approach works using smaller, open-sourced models.

**Reproducibility:**

4: Could mostly reproduce the results, but there may be some variation because of sample variance or minor variations in their interpretation of the protocol or method.

**Reviewer Confidence:**

3: Pretty sure, but there's a chance I missed something. Although I have a good feel for this area in general, I did not carefully check the paper's details, e.g., the math, experimental design, or novelty.

---

> ### Author Rebuttal · Authors · 2023-08-28
>
> Thank you for your very helpful review! We have conducted additional experiments and evaluations, which can be included in the final paper. Please see our response below and feel free to also read through the rebuttals to the other reviews on our paper for more details on performance and rewriting of the Methods section.
>
> ## ad. Reasons to Reject:
> ### 1) Include Cost and runtime
> Yes, we agree. We did not give enough attention to evaluate the run times and costs in the original submission. As a reaction, we have conducted additional experiments and present the total cost as well as the 5-run-average run time (seconds) using different "N" in the N-shot-generator on the WebNLG dataset below (for the parsing error results on WebNLG, refer to rebuttal for review 4Jj1).
>
> #### Runtime costs (WebNLG)
> The below table presents mean runtime cost in seconds from 5 experiment runs.
> As our runs build up upon the results of previous runs (e.g. 1shot run will take already produced results from 0shot run and only rerun the ones, which had parsing errors), we also add a **Cummulative mean time** which adds up the total time it would require to run all shots of the respective experiment from scratch.
> The dataset is **WebNLG**, containing 354 unique relations, meaning that there are in total 354 calls required to the 0x model (zero-shot call to the base LLM).
> Base LLM is GPT3.5-Turbo.
> For simplicity, we use the same model for all shots of the N-shot Generator (1x, 2x, ...) and also for the Consistency Validator.
>
> | **Experiment **  | **Mean time (s)** | **Cummulative mean time (s)** |
> |------------------|:-----------------:|:-----------------------------:|
> | **0x**           |       377.5       |             377.5             |
> | **0x (with CV)** |       173.3       |             550.8             |
> | **1x**           |       142.4       |             519.9             |
> | **2x**           |       118.3       |             638.1             |
> | **3x**           |       159.3       |             797.5             |
> | **4x**           |       111.7       |             909.2             |
> | **5x**           |       108.1       |            1017.2             |
> | **5x (with CV)** |       138.8       |            1156.0             |
>
> #### Estimated API call costs
> In total our costs of API calls were cca 191 USD. However, we did many redundant calls to the API during the development of ASPIRO.
> During our experiments, we mostly utilized the GPT3-davinci model, to keep the results consistent with what (Xiang et al., 2022) reported in their ASDOT paper.
> To get a worst-case-scenario costs of ASPIRO, we calculate below a situation with base  GPT3.5-turbo (cost of $0.002 per 1000 tokens).
>
> Estimated experiments costs for Rel2Text, WebNLG and DART datasets.
> Model: GPT3.5-turbo
>
> | **Experiment**                  | **Total N-SHOT** | **Total CV (G3P5T)** | **Total Cost (G3P5T)** |
> |---------------------------------|:----------------:|:--------------------:|:----------------------:|
> | **REL2TEXT ASDOT_0xG3P5T (V4)** |       0.07       |         0.14         |          0.22          |
> | **REL2TEXT ASDOT_1xG3P5T (V4)** |       0.14       |         0.14         |          0.29          |
> | **REL2TEXT ASDOT_5xG3P5T (V4)** |       0.43       |         0.14         |          0.58          |
> | **REL2TEXT JSON_0xG3P5T (V4)**  |       0.14       |         0.14         |          0.28          |
> | **REL2TEXT JSON_1xG3P5T (V4)**  |       0.27       |         0.14         |          0.42          |
> | **REL2TEXT JSON_5xG3P5T (V4)**  |       0.81       |         0.14         |          0.96          |
> | **WEBNLG ASDOT_0xG3P5T (V4)**   |       0.11       |         0.23         |          0.34          |
> | **WEBNLG ASDOT_1xG3P5T (V4)**   |       0.23       |         0.23         |          0.45          |
> | **WEBNLG ASDOT_5xG3P5T (V4)**   |       0.68       |         0.23         |          0.91          |
> | **WEBNLG JSON_0xG3P5T (V4)**    |       0.21       |         0.23         |          0.44          |
> | **WEBNLG JSON_1xG3P5T (V4)**    |       0.42       |         0.23         |          0.65          |
> | **WEBNLG JSON_5xG3P5T (V4)**    |       1.27       |         0.23         |          1.50          |
> | **DART ASDOT_0xG3P5T (V4)**     |       0.46       |         0.92         |          1.38          |
> | **DART ASDOT_1xG3P5T (V4)**     |       0.92       |         0.92         |          1.84          |
> | **DART ASDOT_5xG3P5T (V4)**     |       2.76       |         0.92         |          3.68          |
> | **DART JSON_0xG3P5T (V4)**      |       0.86       |         0.92         |          1.78          |
> | **DART JSON_1xG3P5T (V4)**      |       1.73       |         0.92         |          2.65          |
> | **DART JSON_5xG3P5T (V4)**      |       5.18       |         0.92         |          6.10          |
>
> Multiply the above table by 10 to get runtime costs for GPT3-davinci model ($0.02 per 1000 tokens).
>
> The table shows, that ASPIRO increases the costs up to 8 to 9-fold if we use 5-shot setting and Consistency Validation. Without Consistency Validation, the maximum cost of **N-shot setting is increased by N-times the initial (zero-shot) cost**.
>
> ### 2) Incremental work
> We believe that the work is valuable as the cost and run time of the conducted experiments are not negligible and results should be useful for future reference in NLG research. We acknowledge that further research on performance of ASPIRO with open source LLMs like Falcon or Llama 2 should be conducted to reduce these costs. Fine-tuning models such as Llama 2 for domain-agnostic data verbalization is also one more logical step to evaluate in further research.
>
> ### 3) Open source models
> We have thought of this option. We did experiments with Falcon LLM 7B during our preliminary research, but we would require additional time and compute power to analyse different open source models. Please see our results on Falcon 7B parsing errors below:
>
> | **Rel2Text parsing errors (total 226 samples)** | **Total Errors** | **Template Errors** | **Missing SUBJECT** | **Multiple SUBJECTs** | **Missing OBJECT** | **Multiple OBJECTs** | **Illegal PLACEHOLDER** |
> |-------------------------------------------------|:----------------:|:-------------------:|:-------------------:|:---------------------:|:------------------:|:--------------------:|:-----------------------:|
> | **a1_FALCON_7B(NONE)(0shot)(NONE)**             |        32        |          28         |          11         |           0           |          2         |          14          |            5            |
> | **a2_FALCON_7B(NONE)(0shot)(FALCON_7B)**        |        30        |          25         |          7          |           1           |          2         |          15          |            5            |
> | **b1_FALCON_7B(FALCON_7B)(1shot)(NONE)**        |        46        |          23         |          23         |           0           |         23         |           0          |            0            |
> | **b2_FALCON_7B(GPT3.5T)(1shot)(NONE)**          |        18        |          18         |          3          |           0           |          1         |          12          |            2            |
>
> You can see that the FALCON_7B model does fairly well in zero-shot setup (a1) with only 28 templates being tagged as having parsing errors. Introducing Consistency Validation to the zero-shot setup (a2) reduces the parsing errors marginally from 28 to 25 templates. Conversely, reprompting FALCON (b1) to try and fix the parsing errors fixes 2 templates, but introduces more errors into the other 23, which is not beneficial.
>
> In b2, we use GPT3.5-Turbo as the retry model, which shows promise in fixing 10 of the templates produced by a1. This shows the possible path of having open source model as "cheap" base model to produce templates and then use larger models such as GPT3.5T to fix the mistakes.
>
> We plan to expand upon this in further work, using Llama2 as well.
>
> ## Note on reproducibility
> We did not mention it in the paper, but we did provide both working code as well as all result data in the supplementary files during the paper submission. We did not publish the code on GitHub as we feared it would clash with the anonymity requirements. We do have working code prepared for GitHub and we will add a reference to it in the final paper version.

---

### Official Review · Reviewer_4Jj1 · 2023-08-05

**Soundness:** 3

**Excitement:**

3: Ambivalent: It has merits (e.g., it reports state-of-the-art results, the idea is nice), but there are key weaknesses (e.g., it describes incremental work), and it can significantly benefit from another round of revision. However, I won't object to accepting it if my co-reviewers champion it.

**Missing References:**

N/A.

**Paper Topic And Main Contributions:**

This paper introduces ASPIRO, a simple yet seemingly effective pipeline for data-to-text generation. ASPIRO consists of two main steps: (i) n-shot example generation and (ii) consistency checking.
* (i) n-shot example generation: In this first stage, a language model generates potential input-output examples, given a test-time input. These examples are then subjected to scrutiny using a rule-based parser to ensure they meet specific predefined criteria.
* (ii) Consistency checking: The second stage involves a consistency validator, which examines the output produced in the first stage. It identifies and addresses potential issues like hallucinations, ambiguities, and structural inconsistencies, aiming to correct them as much as possible.

ASPIRO's effectiveness is evaluated through experiments on the DART and Rel2Text datasets, showing a reduction in parsing errors and relatively high BLEURT scores. Compared to Kasner et al. (2023)’s method, ASPIRO shows competitive performance, though some improvements appear to be incremental. That said, the extent of ASPIRO's generalizability and robustness remains somewhat unclear due to the limited number of experiments conducted by the authors. Further investigations and evaluations across diverse datasets would provide a more comprehensive understanding of ASPIRO's capabilities and limitations.

**Questions For The Authors:**

* Question A: Have you conducted experiments on popular datasets like WebNLG and E2E NLG? If so, how does the proposed ASPIRO pipeline compare to baseline methods in terms of performance?
* Question B: How does the ASPIRO method perform without the second step, i.e., the consistency validator? It would be valuable to include discussions on the significance of the consistency validator and its impact on the overall performance.
* Question C: In Table, does "(Kasner et al., 2023) fewshot-200" mean that Kasner et al. used 200 input-output examples in their few-shot setup?

**Reasons To Accept:**

* Whilst the paper is difficult to follow at times, the motivation and the contributions of the work are clear. The proposed ASPIRO method provides a simple and versatile domain/task-agnostic pipeline for verbalization of data entries to relatively short sentences using a two-stage framework.
* The authors measure the performance of their models along multiple automated metrics, including BLEU, METEOR, semantic similarity score using NUBIA, and PARENT F-1 score. These evaluation measures add more rigor and credibility to the presented findings.

**Reasons To Reject:**

* The main experiments are exclusively conducted on the DART dataset. This limited scope raises concerns about the generalizability and robustness of the proposed method, as it lacks evaluation on other standard data-to-text datasets such as WebNLG (Castro Ferreira et al., 2020) and E2E NLG (Novikova et al., 2017). The absence of such diverse evaluations leaves readers questioning the broader applicability and performance of the ASPIRO approach on various data-to-text scenarios. Including experiments on other datasets would be crucial in addressing these concerns and validating the method's effectiveness across different domains and tasks.
* Section III (Methods), the most critical section of the paper, is challenging to understand and follow. While I understand and acknowledge the constraints of limited space for method description, I believe the authors could have provided a clearer and more thorough explanation of their approach, covering all the involved steps. A comprehensive and well-articulated depiction of the method would have been useful in aiding the reader’s understanding of the contributions of this work.

**Reproducibility:**

3: Could reproduce the results with some difficulty. The settings of parameters are underspecified or subjectively determined; the training/evaluation data are not widely available.

**Reviewer Confidence:**

3: Pretty sure, but there's a chance I missed something. Although I have a good feel for this area in general, I did not carefully check the paper's details, e.g., the math, experimental design, or novelty.

**Typos Grammar Style And Presentation Improvements:**

* Please rewrite Section III to improve its readability and presentation.
* L025: RDF is not defined.
* L037: Zero-shot → zero-shot [lowercase]
* L043: Large Language Model (LLM) → large language model (LLM) [lowercase]
* L101/L102: Change the @ symbol to the section symbol § (it is \S in LaTeX)
* L112: Extra comma after (3.1)
* L124: No need to capitalize Large Language Model (also can use the acronym now)
* Table 2: BLEU and METEOR should be capitalized.
* L238: Pretrained Language Models → pre-trained language models
* Tables 3 and 4: Not so clear what they represent and indicate.

---

> ### Author Rebuttal · Authors · 2023-08-28
>
> Thank you for your thorough review, we have gone through your remarks and wanted to provide some clarifications below:
>
> ## **ad. Reasons To Reject:**
> ### 1) experiments on WebNLG and E2E NLG.
>  - Please see Question A in the ad. Questions section below.
> ### 2) Methods section:
>  - We have rewritten the Methods section to increase readability and completeness, introducing all sub-modules in their own sections and separating the step-by-step explanation in it's own section. Please see it at the end of this rebuttal (it is automatically converted to Markdown from LaTeX, so the formatting may be off in some sections).
>
> ## ad. Questions:
> ### Question A:
> "Have you conducted experiments on popular datasets like WebNLG and E2E NLG? If so, how does the proposed ASPIRO pipeline compare to baseline methods in terms of performance?"
> #### Answer:
> We have originally not done any experiments on WebNLG or E2E NLG. Our reasoning was that DART already contains entries from both WebNLG and E2E. Moreover the E2E dataset only has 7 unique relations (8 if you count familyFriendly yes/no as 2), meaning that there are only 8 templates to produce in each experiment, so statistical evaluation of ASPIRO effectiveness would be difficult with such small sample size.
>
> We also believed that the performance on DART would also be representative of expected performance on WebNLG and E2E, but after running the additional analysis below, it is clear that our assumtion was wrong.
> #### Results on WebNLG
> To analyze performance on WebNLG, we followed a similar approach as with Rel2Text in the paper. We observed both the Parsing Errors and the Automatic Metrics. Please see the parsing errors table below, which
> ##### Parsing Errors
> In the below results, *gpt-3.5-turbo* is used as LLM instances of all calls to both N-shot generator LLM stack and Consistency Validator.
> The table shows number of errors tagged in generated templates by our Rule-based parser for different experiment setups (a1 to f2).
>
> | **Parsing errors, WebNLG dataset (354 total templates)** | **Total Errors** | **Template Errors** | **Missing SUBJECT** | **Multiple SUBJECTs** | **Missing OBJECT** | **Multiple OBJECTs** | **Illegal PLACEHOLDER** |
> |----------------------------------------------------------|:----------------:|:-------------------:|:-------------------:|:---------------------:|:------------------:|:--------------------:|:-----------------------:|
> | **a1_GPT3.5T(NONE)(0shot)(NONE)**                        |        80        |          79         |          24         |           0           |         56         |           0          |            0            |
> | **a2_(NONE)(0shot)(GPT3.5T)**                            |        83        |          79         |          23         |           0           |         57         |           2          |            1            |
> | **b_(GPT3.5T)(1shot)(NONE)**                             |        80        |          79         |          22         |           0           |         58         |           0          |            0            |
> | **c_(GPT3.5T)(2shot)(NONE)**                             |        81        |          80         |          23         |           0           |         58         |           0          |            0            |
> | **d_(GPT3.5T)(3shot)(NONE)**                             |        80        |          79         |          23         |           0           |         57         |           0          |            0            |
> | **e_(GPT3.5T)(4shot)(NONE)**                             |        80        |          79         |          22         |           0           |         58         |           0          |            0            |
> | **f1_(GPT3.5T)(5shot)(NONE)**                            |        80        |          79         |          22         |           0           |         58         |           0          |            0            |
> | **f2_(NONE)(5shot)(GPT3.5T)**                            |        83        |          79         |          23         |           0           |         57         |           2          |            1            |
>
> The table for parsing errors shows that on WebNLG, ASPIRO is generally not able to fix any errors and Consistency Validation actually increases the number of total errors, making 3 of the templates more flawed than without CV.
> ##### Automatic Metrics
> We also compare the templates generated by ASPIRO to the manually crafted templates from Kasner and Dusek, (2022)'s zero-shot Neural pipeline paper to evaluate lexical similarity using PARENT, BLEU and METEOR.
>
> | **Automatic Metrics vs human-crafted templates** | **PARENT_P** | **PARENT_R** | **PARENT_F1** |  **BLEU**  | **METEOR (%)** |
> |--------------------------------------------------|:------------:|:------------:|:-------------:|:----------:|:--------------:|
> | **a1_GPT3.5T(NONE)(0shot)(NONE)**                |    0.7455    |    0.9594    |  **0.8308 **  |   54.50    |   **46.71 **   |
> | **a2_(NONE)(0shot)(GPT3.5T)**                    |    0.7461    |    0.9524    |   _0.8278 _   | **55.00 ** |    _46.20 _    |
> | **b_(GPT3.5T)(1shot)(NONE)**                     |    0.7454    |    0.9591    |    0.8305     |   54.37    |     46.54      |
> | **c_(GPT3.5T)(2shot)(NONE)**                     |    0.7455    |    0.9592    |    0.8306     |   54.44    |     46.56      |
> | **d_(GPT3.5T)(3shot)(NONE)**                     |    0.7456    |    0.9595    |    0.8308     |   54.47    |     46.58      |
> | **e_(GPT3.5T)(4shot)(NONE)**                     |    0.7456    |    0.9595    |    0.8308     |   54.44    |     46.59      |
> | **f1_(GPT3.5T)(5shot)(NONE)**                    |    0.7456    |    0.9595    |    0.8308     |   54.44    |     46.59      |
> | **f2_(NONE)(5shot)(GPT3.5T)**                    |    0.7461    |    0.9537    |   _0.8285 _   | **54.89 ** |    _46.34 _    |
>
> The results are marginal at best and we can only see increase in BLEU score, while PARENT_F1 and METEOR are highest for zero-shot setting.
> (We are in process of calculating the semantic scores \[BLEURT and NUBIA\], but it requires more time.)
> #####  Conclusion
> Contrary to our original belief, we can conclude that ASPIRO pipeline does not perform well on WebNLG and it is important to state that in the final paper. Thank you for pointing this out. We will provide these additional results in the appendix of the final paper.
>
> ### Question B:
> "How does the ASPIRO method perform without the second step, i.e., the consistency validator? It would be valuable to include discussions on the significance of the consistency validator and its impact on the overall performance."
> #### Answer:
> As we mention in the **Operational costs** paragraph of the **Limitations** section, we believe that the consistency validator is not significantly beneficial to ASPIRO performance. We have conducted an ablation study on Rel2Text dataset, but due to the monthly cost hard-cap on OpenAI API it was not very thorough and we did not give it enough focus. Below is the result of our experiments. First 2 rows present GPT3.5Turbo as the base model for all parts of ASPIRO, while rows 3 and 4 use GPT3-davinci as the base model.
>
> Number of parsing error counts without and with (**CV** ) consistency validation:
>
> | **error counts**                     | **Total Errors** | **Template Errors** | **Missing SUBJECT** | **Missing OBJECT** | **Illegal PLACEHOLDER** |
> |--------------------------------------|:----------------:|:-------------------:|:-------------------:|:------------------:|:-----------------------:|
> | **5xGPT3.5Turbo**                    |        12        |          11         |          2          |         10         |            0            |
> | **5xGPT3.5Turbo (CV GPT3.5Turbo)**   |        10        |          9          |          2          |          8         |            0            |
> | **5xGPT3-davinci**                   |        23        |          21         |          5          |         18         |            0            |
> | **5xGPT3-davinci (CV GPT3-davinci)** |        16        |          14         |          3          |         13         |            0            |
>
> Automatic metrics without and with (**CV** ) consistency validation:
>
> | **automatic metrics for Rel2Text test set** | **BLEU** | **METEOR (%)** | **BLEURT** | **NB** | **SS** | **C (%)** | **I (%)** | **E (%)** | **PARENT_F1** |
> |---------------------------------------------|:--------:|:--------------:|:----------:|:------:|:------:|:---------:|:---------:|:---------:|:-------------:|
> | **5xGPT3.5Turbo**                           |   50.88  |      44.97     |   0.8215   | 0.8725 |  4.73  |    3.36   |    8.04   |   88.59   |     0.8910    |
> | **5xGPT3.5Turbo (CV GPT3.5Turbo)**          |   50.89  |      44.95     |   0.8213   | 0.8718 |  4.73  |    3.02   |    8.34   |   88.64   |     0.8908    |
> | **5xGPT3-davinci**                          |   50.96  |      44.69     |   0.8200   | 0.8680 |  4.71  |    4.92   |    7.59   |   87.48   |     0.8879    |
> | **5xGPT3-davinci (CV GPT3-davinci)**        |   51.32  |      44.77     |   0.8200   | 0.8702 |  4.72  |    3.35   |    7.70   |   88.94   |     0.8895    |
>
> The above tables show, that CV is effective at reducing the Parsing error count slightly (12 to 10 and 23 to 16), but only shows marginal gains on statistical metrics.
> We will additionally include these ablation results and the discussion on the importance of Consistency Validation in the appendix.
> ### Question C:
> "In Table, does "(Kasner et al., 2023) fewshot-200" mean that Kasner et al. used 200 input-output examples in their few-shot setup?"
> #### Answer:
> Yes, the number X (in fewshot-X) refers to the number of input-output examples from Rel2Text training set, that were used to fine-tune the BART model before evaluating it on the Rel2Text test dataset. It is stated in the caption of Table 2.
>
> ## Note on Reproducibility
> We did not mention it in the paper, but we did provide both working code as well as all result data in the supplementary files during the paper submission. We did not publish the code on GitHub as we feared it would clash with the anonymity requirements. We do have working code prepared for GitHub and we will add a reference to it in the final paper version.
>
> # Methods (revised)
> The proposed method (ASPIRO) revolves around the conversion of structured data samples into verbalisation templates using a two-stage pipeline: **$N$-shot Generator** (§subsec:n-shot-generator) and **Consistency Validator** (§subsec:consistency-validator). The pipeline processes structured data samples, wherein each sample comprises of one or more RDF triples which share the same relation. ASPIRO (see Figure `fig:ASPIRO-schema`) starts with an initial prompt to verbally articulate the structured data. This is equivalent to prompting a single LLM directly. If the zeroth attempt isn't accurate, it will retry a maximum of $N$ times, refining the previous completion based on parsing errors (§subsec:rule-based-parser). Subsequently, the outputs are validated for consistency, ensuring faithful and reliable verbalisations. We explain the individual stages and their sub-modules in the sections below. Refer to Figure `fig:ASPIRO-schema` for full pipeline and terminology and Figure `fig:ASPIRO-example` for a more illustrative example of ASPIRO pipeline on a single input.
>
> ## $N$-shot Generator
> §subsec:n-shot-generator
>
> $N$-shot Generator further fractures into an LLM stack and a Rule-based parser. The LLM Stack is tasked with generating verbalisation attempts based on given initial prompt (see §subsec:app-prompt-asdot). It does so with the help of the Rule-based parser. This parser checks the generated completions for structural accuracy, ensuring they adhere to expected patterns.
>
> ### LLM Stack
> §subsec:llm-stack
>
> The LLM stack is a set of $N+1$ LLMs, indexed from $0$ to $N$, where $\mathcal{L}_0$ is responsible for the initial completion, whereas each further retry shot initiated by the Rule-based parser (§subsec:rule-based-parser) increments the index by $1$. Each $L_n$ is instantiated separately and does not have to be the same model. Equation (1) shows the single completion for structured input sample $x$ at shot $n`.
>
> $$
> y_n = \mathcal{L}_n(\mathcal{T}(x)) \tag{1}
> $$
>
> where $\mathcal{T}$ is a given prompt and can be either $\mathcal{T}_I$ (initial) or $\mathcal{T}_R$ (retry).
>
> ### Rule-based parser
> §subsec:rule-based-parser
>
> A purely algorithmic module, which validates $y_n$ against a set of conditions $\{\mathcal{C}\}$ one by one. If $y_n$ does not pass the condition $C_i$, a respective parsing error is logged into set $\mathcal{E}_n$. The rules for each given completion are as follows:
>
> $\mathcal{C}_0$ ... has exactly one `<subject>` substring.
>
> $\mathcal{C}_1$ ... has exactly one `<object>` substring.
>
> $\mathcal{C}_2$ ... has no other `<...>` substrings.
>
> If the parser identifies any error in the structure, the next LLM in the LLM stack is re-prompted with Retry Prompt (§subsec:app-prompt-retry) to generate new completion.
>
> ## Consistency Validator
> §subsec:consistency-validator
>
> Even if the outputs from the $N$-shot Generator adhere to the structural patterns, they might still contain inaccuracies, such as hallucinated content. This module assesses the quality of the verbalisations, using the PARENT statistical metric [dhingra-etal-2019-handling]. If PARENT F1 score is too low, the module will utilise an LLM with specialised Consistency Prompt (§subsec:app-prompt-consistency) to improve the sentence.
>
> ### PARENT$_{F1}$ threshold
> §subsec:parent-score
>
> To gauge the quality of the completion $y_n$ from $N$-shot Generator, we set a minimal threshold ($\mu$) for the PARENT score of $y_n$. The score is calculated using eq. (3) against artificially constructed table and reference.
>
> First, we construct the respective hypothesis, table and reference entries:
>
> $$ h = y_n\texttt{.replace}([s, o], e) \\ t = \langle e, r\texttt{.split(" ")}, e \rangle \\ \rho = r \tag{2}
> $$
>
> where `<subject>` and `<object>` are replaced with `<entity>` to prevent penalizing order discrepancy between hypothesis and table.
>
> We then calculate the PARENT F1 score using equation (3).
>
> $$
> F1(y_n) = \texttt{PARENT}(h, \rho, t) \tag{3}
> $$
>
> ### Consistency LLM
> §subsec:consistency-llm
>
> If the calculated PARENT score from §subsec:parent-score is not sufficient, we call another LLM with prompt $\mathcal{T}_C$ as in eq. (4).
>
> $$
> y_C = \mathcal{L}_C(\mathcal{T}_C(r, y_n)) \tag{4}
> $$
>
> The prompt $\mathcal{T}_C$ is designed to guide $\mathcal{L}_C$ to identify problems with the given completion, provide advice how to fix it and subsequently produce fixed completion in a structured json output. See §subsec:app-prompt-consistency for full version of the prompt.
>
> ## Detailed Pipeline Formulation
> §subsec:formulation
>
> Given a dataset of structured data samples $\{x^r\}_{r\in\mathcal{R}}$, where $x^r=\{x^r_1, x^r_2, ..., x^r_m\}$ and $x^r_j$ is a single RDF triple $x^r_j=\langle s^r_j, r, o^r_j\rangle$ with relation $r\in \mathcal{R}$, the pipeline for one $x^r$ is as follows:
>
> **Step 0** Set $n=0$ and $\mathcal{T}^r_0=\mathcal{T}_I(x^r)$.
>
> **Step 1** Calculate $y^r_n$ using eq. (1).
>
> **Step 2** Use §subsec:rule-based-parser to validate $y^r_n$ against all conditions ${\mathcal{C}}$. If errors ($\mathcal{E}^r_n$)  are found, run equation (5) and return to **Step 1**. Otherwise go to **Step 3**.
>
> $$
>     \mathcal{T}^r_{n+1} = \mathcal{T}_R(x^r, y^r_n, \mathcal{E}^r_n) \\
>     n = n+1 \tag{5}
> $$
>
> **Step 3** Use §subsec:parent-score and calculate $F1(y^r_n)$ via eq. (3). If the calculated $F1$ score is lower than our chosen threshold $0\leq\mu \leq 1$, continue to **Step 4**. Otherwise, output current $y_n^r$ as the final completion $y^r$.
>
> **Step 4** Use §subsec:consistency-llm with prompt $\mathcal{T}_C^r$ and calculate $y^r_C$ via eq. (4).
>
> **Step 5** Compute \(F1\) scores of \(y_n^r\) and \(y_C^r\) using eq. (3) and take the completion with higher score via eq. (6) to produce the final completion
> $$
>     y^r = \underset{y\in\{y_n^r, y_C^r\}}{\texttt{argmax}(F1(y))} \tag{6}
> $$

---

### Meta-Review · Area_Chair_1Di2 · 2023-09-17

**Recommendation:** 3

**Metareview:**

This study presents an approach for verbalizing structured data into concise template sentences, achieved by instructing Large Language Models (LLMs) to generate entity-agnostic templates while incorporating additional algorithmic parsing consistency checks.

**Pros**:
- The motivation behind the work and its contributions are clearly articulated.
- The proposed method yields significant performance improvements in data-to-text generation tasks.
- The authors use a diverse array of automated metrics for evaluation, enhancing the rigor and credibility of their results.

**Cons**:
- Two reviewers note that the paper's complexity is exacerbated by an abundance of mathematical expressions, which can obscure the paper's core message.
- The method is complex and requires numerous LLM API calls.

---

### Decision · Program_Chairs · 2023-10-07

**Decision:**

Accept-Findings

**Comment:**

This study presents an approach for verbalizing structured data into concise template sentences, achieved by instructing Large Language Models (LLMs) to generate entity-agnostic templates while incorporating additional algorithmic parsing consistency checks.

**Pros**:
- The motivation behind the work and its contributions are clearly articulated.
- The proposed method yields significant performance improvements in data-to-text generation tasks.
- The authors use a diverse array of automated metrics for evaluation, enhancing the rigor and credibility of their results.

**Cons**:
- Two reviewers note that the paper's complexity is exacerbated by an abundance of mathematical expressions, which can obscure the paper's core message.
- The method is complex and requires numerous LLM API calls.